# Atom Probe Tomography of Aluminium Alloys: A Systematic Meta-Analysis Review of 2018

**Anna V. Ceguerra [1],\* and Ross K.W. Marceau [2]**

1   Australian Centre for Microscopy & Microanalysis, and School of Aerospace, Mechanical and Mechatronic Engineering, The University of Sydney, Sydney, NSW 2006, Australia
2   Institute for Frontier Materials, Deakin University, Geelong, VIC 3216, Australia; ross.marceau@deakin.edu.au
\*   Correspondence: anna.ceguerra@sydney.edu.au; Tel.: +61-2-9036-6416

**Abstract:** Atom probe tomography (APT) is a microscopy technique that provides a unique combination of information, specifically the position and elemental identity of each atom in three dimensions. Although the mass and spatial resolution is not perfect, we are still able to gain insights into materials science questions that we cannot access using other techniques. This systematic meta-analysis review summarises research in 2018 that used APT to study materials science questions in aluminium alloys.

**Keywords:** atom probe tomography; aluminium alloys; structure–property relationships; meta-analysis

---

## 1. Introduction

Atom probe tomography is a powerful technique that allows researchers to gain insights into a material at the atomic scale and in three dimensions in a way that is not possible using any other microscopy method. Despite known imperfections with the absolute atomic accuracy of the reconstructed data, atom probe tomography can still be used to answer specific types of questions.

In this work, we performed a meta-analysis of 34 original research articles of aluminium alloys published in 2018 [1–34] to determine (1) the types of materials science questions being studied that use atom probe tomography, (2) how atom probe tomography is being used to gather insights into these questions, and (3) how researchers handle the imperfect nature of the data.

## 2. Methods

The Preferred Reporting Items for Systematic Reviews and Meta-Analyses (PRISMA) method [35] was used to determine the data for this systematic review. We used the tools XMind, Zotero and Microsoft Excel to perform the meta-analysis. Once the set of papers was determined, the full text was read and a list of tags relating to atom probe tomography were associated with each paper.

## 3. Results

### 3.1. PRISMA Flow

In the identification stage, 442 papers were found using Scopus by searching for "atom probe" in quotation marks within the title, keywords and abstract. This list of records was downloaded as an RIS file and imported into both EndNote and Zotero.

In the screening stage, the titles of 442 records were exported as a list of references from EndNote into Microsoft Word. The author and titles were then imported into XMind. Each title was classified to develop the initial list of themes and subthemes. Then, the 442 records were tagged more specifically

on Zotero based on titles and according to the themes. During this process one article was retracted and was therefore excluded from the next step.

In the eligibility stage, the full text of 38 articles was accessed from the list of 441 papers. These articles were chosen as they were identified with the "Metal-Al" tag prefix. Four articles were excluded based on the full text, where two were review articles, one was modelling work, and one was written in Japanese with no translation. The full text of a paper was associated with a list of structured keywords, prefaced with the text "FT- ". The Supplementary Material includes the extracted RIS database of this step.

The result is that the full texts of 34 articles [1–34] were analysed for both qualitative and quantitative analyses. Appendix A contains the number of instances associated with each tag, grouped according to topic and subtopics.

### 3.2. Meta-Analysis—Materials Science Questions

For this meta-analysis, most papers had only one type of materials science relationship question. There were two exceptions that had no relationship marked (one due to an error, and one only investigated structure). Five papers also had multiple relationship types because they investigated multiple questions.

The information from each table was extracted from the materials science question tags from Appendix A. This was done by counting the instances for each tag that summarised the questions from the paper. We identified types of:

- Materials science relationship questions being asked (Table 1) from the tags in Appendix A.3.1;
- Materials science phenomena identified (Table 2) from the tags in Appendix A.3.2;
- Processing applied to the samples (Table 3) from the tags in Appendix A.3.1;
- Structure being studied in the samples (Table 4) from the tags in Appendix A.3.5;
- Properties associated with the samples (Table 5) from the tags in Appendix A.3.6.

**Table 1.** Types of materials science relationship questions studied by atom probe tomography (APT).

| Question Type | Instances |
|---|---|
| Composition–Processing–Structure–Property | 18 |
| Composition–Processing–Structure | 1 |
| Processing–Structure–Property | 9 |
| Composition–Structure | 1 |
| Processing–Structure | 11 |
| Structure–Property | 1 |

**Table 2.** Types of materials science phenomena studied by APT.

| Phenomenon | Instances |
|---|---|
| Theory | 7 |
| Composition | 3 |
| Processing | 11 |
| Structure—Lattice | 2 |
| Structure—Dislocation | 3 |
| Structure—Particles | 26 |
| Structure—Grain | 5 |
| Property | 5 |

**Table 3.** Types of processing applied to samples studied by APT.

| Type | Instances |
|------|-----------|
| Aging | 24 |
| HIP [1] | 2 |
| other | 4 |

[1] Hot isostatic pressing.

**Table 4.** Types of structure investigated using APT.

| Type | Instances |
|------|-----------|
| Solid Solution | 9 |
| Defect | 1 |
| Cluster | 25 |
| Precipitate | 41 |
| Boundary | 9 |
| Microstructure | 6 |

**Table 5.** Types of properties investigated.

| Type | Instances |
|------|-----------|
| Creep | 5 |
| Functional | 6 |
| Corrosion | 2 |
| Hardness | 26 |

*3.3. Meta-Analysis—Atom Probe Tomography (APT)*

For this meta-analysis, the information from each table was extracted from the APT tags in Appendix A. Multiple analysis types were identified in each paper. This was done by counting the instances for each tag that summarised the questions from the paper. We identified types of:

- APT analysis (Table 6) from the tags in Appendices A.1.1–A.1.5;
- APT artefact handling (Table 7) from tags in Appendix A.1.6.

**Table 6.** The number of instances of each type of APT analysis.

| Type | Instances | Papers |
|------|-----------|--------|
| Composition | 30 | 19 |
| Cluster | 17 | 11 |
| Neighbourhood | 5 | 4 |
| Visualisation | 45 | 33 |
| Particle features | 41 | 16 |

**Table 7.** The number of instances of handling APT artefacts.

| Type | Instances |
|------|-----------|
| Mass Spectrum | 5 |
| Particle Concentration | 3 |
| Poles | 3 |
| Preferential Evaporation | 2 |
| Sampling | 1 |

### 3.4. Meta-Analysis—Linking APT Instrument Type with Materials Science Question

In this analysis, the instrument type was linked with the number tags for each type of materials science question (Table 8). Each tag had multiple papers, but we reported the number of tags rather than the number of papers.

**Table 8.** Instrument versus question type (number of tags).

| Type | CPSP | CPS | PSP | CS | PS | SP | TOTAL |
|------|------|-----|-----|----|----|----|-------|
| Laser | 6 | 1 | 6 | 1 | 8 | 0 | **22** |
| Voltage | 11 | 0 | 0 | 0 | 3 | 1 | **15** |
| Straight | 6 | 0 | 3 | 0 | 2 | 0 | **11** |
| Reflectron | 11 | 1 | 3 | 1 | 9 | 1 | **26** |
| **TOTAL** | **34** | **2** | **12** | **2** | **22** | **2** | |

CPSP: composition–processing–structure–property (in that order); CPS: composition–processing–structure; PSP: processing–structure–property; CS: composition–structure; PS: processing–structure; SP: structure–property.

### 3.5. Meta-Analysis—Linking APT Analysis Type with Materials Science Question

In this analysis, the APT analysis type was linked with the number of tags for each type of materials science question (Table 9). Similar to the previous section, each tag had multiple papers, but we reported the number of tags rather than the number of papers.

**Table 9.** APT analysis type versus question type (number of tags).

| Type | CPSP | CPS | PSP | CS | PS | SP | TOTAL |
|------|------|-----|-----|----|----|----|-------|
| Composition | 7 | 1 | 1 | 1 | 5 | 6 | **21** |
| Cluster | 6 | 0 | 0 | 0 | 3 | 5 | **14** |
| Neighbourhood | 2 | 0 | 0 | 0 | 1 | 5 | **8** |
| Particle Features | 6 | 0 | 0 | 0 | 6 | 6 | **18** |
| Artefacts | 4 | 1 | 0 | 0 | 5 | 2 | **12** |
| Data Quality | 3 | 0 | 0 | 0 | 1 | 0 | **4** |
| **TOTAL** | **28** | **2** | **1** | **1** | **21** | **24** | |

## 4. Discussion

We determined responses to the following questions based on the data.

### 4.1. What Types of Materials Science Questions Do We Gain Insight into by Using APT?

The top three materials science relationship questions investigated using APT—comprising 92.7% of instances—involved processing and structure. These instances include composition–processing–structure–property relationships (43.9%), processing–structure relationships (26.8%) and processing–structure–property relationships (22.0%).

The top three materials science phenomena questions investigated using APT were the structure of particles (41.9%), the effect of processing conditions on properties (17.7%), and theories describing the phenomena (11.29%). Of the phenomena identified, 58% were related to the structure of the material. The full list of specific keywords for each type of question are listed in Appendix A.3.2.

The most popular types of structure investigated using APT were precipitates (45.1% of instances) and clusters (27.5% of instances). Boundaries and solid solution came in at an equal third. This is consistent with the strengths of APT in investigating 3D structure that cannot be observed using any other means.

Hardness was the most popular property correlated with the aluminium alloy structural investigations using APT, comprising 66.7% of total instances.

*4.2. Is Data Quality Identified as a Significant Issue?*

In terms of data quality, 11 (32.4%) of the 34 papers highlighted specific APT artefacts and how they considered this in the interpretation of their data. The other papers did not see this as a significant concern in their particular datasets. Specific issues mentioned were:

- Composition differences between what was observed in APT and what was expected, in either the bulk or in particles;
- Overlapping peaks in the mass spectrum, causing misidentification of some proportion of the ions;
- Trajectory aberrations or solute segregation at the poles;
- Preferential field evaporation of specific elements;
- Limitations of the sampling with the limited analysis volume.

Five papers (14.7%) mentioned how they improved the data quality of the APT data through removal of regions with artefacts such as poles or chemical segregation. Six papers included information on how they calibrated the spatial reconstruction, and two of those also included consideration of the mass-to-charge-ratio ranging issues.

*4.3. Is There a Relationship between Instrument Type and the Materials Science Question Being Asked?*

In 33 instances a type of local electrode atom probe (LEAP) was identified and in one instance a tomographic atom probe (TAP) was used. For laser versus voltage machines, 60.6% were identified as laser instruments and 36.4% were voltage-only instruments. For the flight path type, 39.4% were identified as straight flight path instruments, while 57.6% were identified as reflectron instruments.

In terms of the type of materials science question being asked, for composition–processing–structure–property (CPSP) questions there was a tendency to use voltage and reflectron flight path instruments (64%). Processing–structure–property (PSP) questions used laser instruments, but there was an even split between straight versus reflectron flight path. Processing–structure (PS) questions tended to use laser and reflectron instruments. The distribution of laser/voltage and reflectron/straight flight path combinations are not publicly available, so we cannot draw conclusions based on the current distribution of instruments.

Note that this study did not effectively capture which mode was being used (either laser or voltage), only what type of instrument was being used. It is anecdotally known that there are trade-offs regarding which modes are used. For example, laser mode improves the yield compared to voltage mode, but voltage mode enhances spatial resolution compared to laser mode. Another example is that reflectron flight path has better mass resolution compared to straight flight path, but straight flight path has better detection efficiency. However, we could not make conclusions on this based on this data.

*4.4. Is the Software Being Used Up-To-Date?*

The current version of IVAS in 2018 was 3.8.x. The most popular software version was IVAS 3.6.12. Of the papers analysed, 68% indicated that they used one of IVAS version 3.6.6 to version 3.8. Matlab is an emerging software platform, whereas PoSAP is an older software platform.

*4.5. What Specimen Preparation Techniques Are Being Used?*

Electrochemical polishing was used in 66.7% studies, while in 22.2% of the studies the gallium focused ion beam (FIB) was used for site-specific specimen preparation. Xenon-based FIB is an emerging technology, and the ElectroPointer is an older technology.

*4.6. What Other Microscopy Techniques Were Used to Support APT Findings?*

There were 20 papers that also used TEM out of the 23 papers that used another form of microscopy to inform their study. Other techniques being used were EBSD (five instances) and SEM (three instances).

*4.7. Comments on the Methodology*

The search criteria limited the papers to only those that had APT as a major component in their investigation and were likely to include original research. We deliberately did not include other papers that merely mentioned APT.

The keyword aspect of publications could be enhanced to better enable this type of study. While some aspects of the study were automated (e.g., generating a raw report of the tags), many aspects—including determining, categorising and synthesising the tags—involved a highly manual process. In tagging, both the title for 442 papers and the full text for 38 papers needed to be read by a human. Categorising involved sifting through keywords, grouping them together into themes and having a feedback process to inform the next iteration, all of which were performed by a human. In synthesising, some of the process was semi-automated, such as calculating the number of papers for a set of tags. This process cannot be handled by current information-processing algorithms such as word clouds or natural language processing. This is because the tags were sometimes determined from a single instance of a particular word, and the tag may not match the actual words as it was published. We note that the ~three keywords per article were not very useful and suggest having an "Expanded Keywords" option to more comprehensively capture the content of the article according to the authors' understanding of the work. We suggest that expansion of the keywords include categorisation of keywords according to the topic area, a more structured way of naming the keywords as shown in this study and an increase in the number of keywords to better represent the content of the paper.

Software currently does not exist that can perform this type of analysis from start to finish. We investigated EndNote, Papers3, NVivo, XMind and Zotero. In the end, we settled for a combination of XMind v8, Zotero v5.0.73 and Excel v16.28 to perform the PRISM method and the resulting meta-analysis.

## 5. Conclusions

This application of meta-analysis for systematic review of APT research of aluminium alloys is in its infancy. However, it is envisaged that the methodology could be valuable in general for the identification of information and trends that could otherwise be difficult to determine with statistical rigour from individual studies. This new information may not only exist within the scientific literature, but could be extended by application of the methodology to APT data repositories or media platforms through which an ever-increasing amount of knowledge and information is being shared.

**Supplementary Materials:** The following are available online at http://www.mdpi.com/2075-4701/9/10/1071/s1. Endnote database: APTofAl-2018.ris.

**Author Contributions:** Conceptualisation, data interpretation and manuscript writing, A.V.C. and R.K.W.M.; PRISM data analysis, A.V.C.

**Funding:** This research received no external funding.

**Acknowledgments:** The authors acknowledge the facilities, scientific and technical assistance, of the University of Sydney node of Microscopy Australia (Sydney Microscopy and Microanalysis).

**Conflicts of Interest:** The authors declare no conflict of interest.

## Appendix A

This appendix contains the full list of tags (grouped according to categories) along with the number of papers having been identified with that tag. "FT" stands for "full text", indicating that this tag was identified from the full text of the article.

*Appendix A.1 APT*

Appendix A.1.1 Composition Analysis

| | |
|---|---|
| FT-APT-analysis-1DcompositionProfile | 6 |
| FT-APT-analysis-2DcompositionProfile | 1 |
| FT-APT-analysis-bulkComposition | 2 |
| FT-APT-analysis-chargeState | 1 |
| FT-APT-analysis-composition-boundarySegregation | 1 |
| FT-APT-analysis-composition-depletion | 1 |
| FT-APT-analysis-composition-dislocation | 1 |
| FT-APT-analysis-composition-enrichment | 1 |
| FT-APT-analysis-composition-matrix | 1 |
| FT-APT-analysis-composition-precipitate | 2 |
| FT-APT-analysis-composition-proxigram | 10 |
| FT-APT-analysis-composition-segregation | 1 |
| FT-APT-analysis-erosionProfile | 1 |
| FT-APT-analysis-isotopes | 1 |

Appendix A.1.2 Cluster Analysis

| | |
|---|---|
| FT-APT-analysis-clustering-classification | 3 |
| FT-APT-analysis-clustering-Felfer | 1 |
| FT-APT-analysis-clustering-maxSep | 8 |
| FT-APT-analysis-clustering-parameterSelection | 4 |
| FT-APT-analysis-clustering-twoStage | 1 |

Appendix A.1.3 Neighbourhood Analysis

| | |
|---|---|
| FT-APT-analysis-correlationFunction-developed | 1 |
| FT-APT-analysis-pairCorrelationFunction-defined | 1 |
| FT-APT-analysis-radialDistributionAnalysis | 1 |
| FT-APT-analysis-radialDistributionFunction | 1 |
| FT-APT-analysis-solute-nearestNeighbour | 1 |

Appendix A.1.4 Visualisation

| | |
|---|---|
| FT-APT-analysis-isosurface-concentration | 12 |
| FT-APT-analysis-isosurface-concentration-parameterSelection | 1 |
| FT-APT-analysis-visualisation-3D | 32 |

Appendix A.1.5 Particle Features

| | |
|---|---|
| FT-APT-analysis-particle-composition | 6 |
| FT-APT-analysis-particle-composition-Al | 1 |
| FT-APT-analysis-particle-composition-GPMRouen | 1 |
| FT-APT-analysis-particle-compositionFraction | 1 |
| FT-APT-analysis-particle-density | 2 |
| FT-APT-analysis-particle-grouped-bySize | 1 |
| FT-APT-analysis-particle-GuinierRadius | 1 |
| FT-APT-analysis-particle-morphology | 1 |
| FT-APT-analysis-particle-numberDensity | 6 |
| FT-APT-analysis-particle-orientation | 1 |
| FT-APT-analysis-particle-radius | 3 |
| FT-APT-analysis-particle-ratio-Cu/Mg | 1 |
| FT-APT-analysis-particle-ratio-Cu/MgSi | 1 |

| | |
|---|---|
| FT-APT-analysis-particle-ratio-CuSi/Mg | 1 |
| FT-APT-analysis-particle-ratio-Mg/MgSi | 1 |
| FT-APT-analysis-particle-ratio-Mg/Si | 4 |
| FT-APT-analysis-particle-ratio-Si/Mg | 2 |
| FT-APT-analysis-particle-ratio-Zn/Mg | 1 |
| FT-APT-analysis-particle-size | 2 |
| FT-APT-analysis-particle-size-soluteAtoms | 1 |
| FT-APT-analysis-particle-volumeFraction | 3 |

## Appendix A.1.6 Artefacts

| | |
|---|---|
| FT-APT-artefact-massSpec-compositionDifference | 1 |
| FT-APT-artefact-massSpec-compositionDifference-O | 1 |
| FT-APT-artefact-massSpec-overlappingPeak-Ti/Mg | 2 |
| FT-APT-artefact-massSpec-overlappingPeak-Zr/Sc | 1 |
| FT-APT-artefact-particleConcentration | 1 |
| FT-APT-artefact-particleConcentration-Al | 2 |
| FT-APT-artefact-poles-segregation-Cu | 1 |
| FT-APT-artefact-poles-trajectoryAberration | 2 |
| FT-APT-artefact-preferentialFieldEvaporation | 1 |
| FT-APT-artefact-preferentialFieldEvaporation-solutes | 1 |
| FT-APT-artefact-sampling-analysisVolume | 1 |

## Appendix A.1.7 Data Quality

| | |
|---|---|
| FT-APT-dataQuality-bulkComposition-siteSpecific | 1 |
| FT-APT-dataQuality-poleRemoval | 2 |
| FT-APT-dataQuality-removal-pole | 2 |
| FT-APT-dataQuality-removal-segregation | 1 |

## Appendix A.1.8 Experiment

| | |
|---|---|
| FT-APT-exp-laser | 1 |
| FT-APT-exp-laser-green | 1 |
| FT-APT-exp-voltage | 2 |

## Appendix A.1.9 Instrument Model

| | |
|---|---|
| FT-APT-instrument-LEAP | 1 |
| FT-APT-instrument-LEAP3000HR | 4 |
| FT-APT-instrument-LEAP3000Si | 1 |
| FT-APT-instrument-LEAP3000XHR | 2 |
| FT-APT-instrument-LEAP4000HR | 7 |
| FT-APT-instrument-LEAP4000XHR | 5 |
| FT-APT-instrument-LEAP4000XSi | 8 |
| FT-APT-Instrument-LEAP5000XR | 1 |
| FT-APT-instrument-LEAP5000XS | 4 |
| FT-APT-instrument-TAP | 1 |

## Appendix A.1.10 Reconstruction

| | |
|---|---|
| FT-APT-reconstruction-ranging-decisions | 1 |
| FT-APT-reconstruction-ranging-manual | 1 |
| FT-APT-reconstruction-spatial-calibration-crystal | 4 |
| FT-APT-reconstruction-spatial-calibration-radiusSEM | 1 |
| FT-APT-reconstruction-verification-volume | 1 |

Appendix A.1.11 Software

| | |
|---|---|
| FT-APT-software-IVAS | 3 |
| FT-APT-software-IVAS3.6 | 1 |
| FT-APT-software-IVAS3.6.0 | 1 |
| FT-APT-software-IVAS3.6.1 | 1 |
| FT-APT-software-IVAS3.6.10 | 1 |
| FT-APT-software-IVAS3.6.12 | 8 |
| FT-APT-software-IVAS3.6.14 | 2 |
| FT-APT-software-IVAS3.6.6 | 2 |
| FT-APT-software-IVAS3.6.8 | 3 |
| FT-APT-software-IVAS3.8.0 | 1 |
| FT-APT-software-matlab | 1 |
| FT-APT-software-PoSAP1.6 | 1 |
| FT-APT-analysis-particleStatisticTool | 1 |

Appendix A.1.12 Specimen Preparation

| | |
|---|---|
| FT-APT-specPrep-electrochemicalPolishing | 18 |
| FT-APT-specPrep-electrochemicalPolishing-ElectroPointer | 1 |
| FT-APT-specPrep-FIB-Ga | 6 |
| FT-APT-specPrep-FIB-Xe | 1 |
| FT-APT-specPrep-transferInInertGas | 1 |

Appendix A.1.13 Random Comparator

| | |
|---|---|
| FT-APT-analysis-randomLabelling | 1 |

*Appendix A.2 Related Techniques*

| | |
|---|---|
| FT-Microscopy-ACTEM | 1 |
| FT-Microscopy-AFM | 1 |
| FT-Microscopy-DSC | 1 |
| FT-Microscopy-EBSD | 5 |
| FT-Microscopy-EDX | 1 |
| FT-Microscopy-HAADFSTEM | 1 |
| FT-Microscopy-HRTEM | 3 |
| FT-Microscopy-LOM | 1 |
| FT-Microscopy-PALS | 1 |
| FT-Microscopy-SANS | 1 |
| FT-Microscopy-SEM | 4 |
| FT-Microscopy-STEM | 3 |
| FT-Microscopy-TEM | 14 |
| FT-Microscopy-TKD | 1 |
| FT-Modelling-DFT | 1 |
| FT-Modelling-firstPrinciples-VASP | 2 |

*Appendix A.3 Materials Science*

Appendix A.3.1 Questions

| | |
|---|---|
| FT-MSE-CPSPR-addition-aging-cluster-hardness | 1 |
| FT-MSE-CPSPR-addition-aging-cluster-yieldStrength | 1 |
| FT-MSE-CPSPR-addition-aging-microstructure-stability | 1 |
| FT-MSE-CPSPR-addition-aging-precipitate-creep | 1 |
| FT-MSE-CPSPR-addition-aging-precipitate-electricalConductivity | 1 |
| FT-MSE-CPSPR-addition-aging-precipitate-hardness | 5 |

| | |
|---|---|
| FT-MSE-CPSPR-addition-aging-precipitate-tensile | 2 |
| FT-MSE-CPSPR-addition-aging-precipitate-yieldStrength | 1 |
| FT-MSE-CPSPR-addition-aging-precipitation-hardness | 1 |
| FT-MSE-CPSPR-addition-aging-solutePartitioning-ductileFracture | 1 |
| FT-MSE-CPSPR-composition-aging-cluster-hardness | 1 |
| FT-MSE-CPSPR-composition-aging-cluster-strength | 1 |
| FT-MSE-CPSPR-composition-aging-precipitate-hardness | 1 |
| FT-MSE-CSR-addition-segregation | 1 |
| FT-MSE-CPSR-addition-aging-solutePartitioning | 1 |
| FT-MSE-PSP-drawing-clusterMorphology | 1 |
| FT-MSE-PSPR-aging-cluster-hardness | 2 |
| FT-MSE-PSPR-aging-microstructure-fractureToughness | 1 |
| FT-MSE-PSPR-aging-microstructure-tensile | 1 |
| FT-MSE-PSPR-aging-precipitate-hardness | 2 |
| FT-MSE-PSPR-aging-soluteDistribution-hardness | 1 |
| FT-PSPR-aging-precipitate-hardness | 1 |
| FT-MSE-PSR-aging-grain | 1 |
| FT-MSE-PSR-aging-precipitate | 2 |
| FT-MSE-PSR-aging-soluteAggregate | 1 |
| FT-MSE-PSR-HIP-grain | 1 |
| FT-MSE-PSR-HIP-precipitate | 1 |
| FT-MSE-PSR-irradiationTemperature-soluteSegregationDislocation | 1 |
| FT-MSE-PSR-magneticAnnealing-precipitate | 1 |
| FT-MSE-PSR-solutionisation-segregation | 1 |
| FT-PSR-aging-precipitate | 2 |
| FT-MSE-SPR-soluteAggregate-hardness | 1 |

## Appendix A.3.2 Phenomena

### Theory

| | |
|---|---|
| FT-MSE-phenomenon-GibbsThomsonEffect | 1 |
| FT-MSE-phenomenon-kinetics | 1 |
| FT-MSE-thermodynamics-interfacialEnergy | 1 |
| FT-MSE-thermodynamics-interfacialExcess | 1 |
| FT-MSE-thermodynamics-phaseDiagram-FactSage | 1 |
| FT-MSE-thermodynamics-precipitationActivationEnergy | 1 |
| FT-MSE-timeTemperaturePrecipitationDiagram | 1 |

### Composition

| | |
|---|---|
| FT-MSE-phenomenon-compositionEvolution | 1 |
| FT-MSE-phenomenon-evolution-phaseChemistry | 1 |
| FT-MSE-phenomenon-soluteRedistribution | 1 |

### Processing

| | |
|---|---|
| FT-MSE-phenomenon-ageHardening | 2 |
| FT-MSE-phenomenon-agingKinetics-hardness | 1 |
| FT-MSE-phenomenon-agingKinetics-TEP | 1 |
| FT-MSE-phenomenon-naturalAging-inhibit | 1 |
| FT-MSE-phenomenon-overAging | 3 |
| FT-MSE-phenomenon-peakAging | 2 |
| FT-MSE-phenomenon-response-bakeHardening | 1 |

### Structure—Lattice

| | |
|---|---|
| FT-MSE-phenomenon-strengthening-lattice | 1 |
| FT-MSE-phenomenon-strengthening-solidSolution | 1 |

### Structure—Dislocation

| | |
|---|---|
| FT-MSE-phenomenon-dislocation-creep | 1 |
| FT-MSE-phenomenon-dislocationHardening | 1 |
| FT-MSE-phenomenon-strengthening-dislocation | 1 |

Structure—Particles

| | |
|---|---|
| FT-MSE-phenomenon-dispersionHardening | 1 |
| FT-MSE-phenomenon-dispersionHardening-AshbyOrowan | 1 |
| FT-MSE-phenomenon-evolution-cluster | 2 |
| FT-MSE-phenomenon-evolution-precipitate | 2 |
| FT-MSE-phenomenon-nucleation | 1 |
| FT-MSE-phenomenon-diffusion-precipitate-shell | 1 |
| FT-MSE-phenomenon-kinetics | 1 |
| FT-MSE-phenomenon-precipitate-coarseningResistance | 1 |
| FT-MSE-phenomenon-precipitate-solutePartitioning | 1 |
| FT-MSE-phenomenon-precipitateDissolve-Al3Er | 1 |
| FT-MSE-phenomenon-precipitateDissolve-Al3Sc | 1 |
| FT-MSE-phenomenon-precipitateDissolve-ErRich | 1 |
| FT-MSE-phenomenon-precipitateHardening | 2 |
| FT-MSE-phenomenon-precipitateOrdering | 1 |
| FT-MSE-phenomenon-precipitateStrengthening | 1 |
| FT-MSE-phenomenon-precipitateTransformation | 1 |
| FT-MSE-phenomenon-precipitation-sequential | 1 |
| FT-MSE-phenomenon-precipitationSequence | 4 |
| FT-MSE-phenomenon-stability-cluster-define | 1 |
| FT-MSE-phenomenon-strengthening-particle | 1 |

Structure—Grain

| | |
|---|---|
| FT-MSE-phenomenon-diffusion-subGrainBoundary | 1 |
| FT-MSE-phenomenon-grainBoundaryStrengthening | 1 |
| FT-MSE-phenomenon-recrystallisation | 1 |
| FT-MSE-phenomenon-twoStageDoublePeaks | 1 |
| FT-MSE-phenomenon-underAged | 1 |

Property

| | |
|---|---|
| FT-MSE-phenomenon-creepResistance | 1 |
| FT-MSE-phenomenon-evolution-microhardness | 1 |
| FT-MSE-phenomenon-diffusion-creep | 1 |
| FT-MSE-phenomenon-strain-creep | 1 |
| FT-MSE-phenomenon-thermalStability | 1 |

Appendix A.3.3 Composition

| | |
|---|---|
| FT-MSE-composition-addition-Ag | 3 |
| FT-MSE-composition-addition-Cd | 2 |
| FT-MSE-composition-addition-Cu | 3 |
| FT-MSE-composition-addition-In | 1 |
| FT-MSE-composition-addition-Mg | 1 |
| FT-MSE-composition-addition-Mn | 1 |
| FT-MSE-composition-addition-Nb | 1 |
| FT-MSE-composition-addition-Ni | 1 |
| FT-MSE-composition-addition-Si | 1 |
| FT-MSE-composition-addition-Sn | 1 |
| FT-MSE-composition-addition-Ta | 1 |
| FT-MSE-composition-addition-Ti | 1 |
| FT-MSE-composition-addition-V | 1 |
| FT-MSE-composition-addition-Y | 1 |
| FT-MSE-composition-addition-Zn | 2 |
| FT-MSE-composition-addition-Zr | 2 |
| FT-MSE-composition-bulk-ratio-Mg/Si | 1 |
| FT-MSE-composition-soluteInteractions | 2 |

## Appendix A.3.4 Processing

| | |
|---|---|
| FT-MSE-environment-highTemp | 2 |
| FT-MSE-processing-aging | 1 |
| FT-MSE-processing-aging-artificial | 22 |
| FT-MSE-processing-aging-cryoHalted | 1 |
| FT-MSE-processing-aging-double | 2 |
| FT-MSE-processing-aging-interrupted | 1 |
| FT-MSE-processing-aging-isochronal | 1 |
| FT-MSE-processing-aging-isothermal | 2 |
| FT-MSE-processing-aging-natural | 5 |
| FT-MSE-processing-annealed | 1 |
| FT-MSE-processing-arcMelting | 2 |
| FT-MSE-processing-asCast | 1 |
| FT-MSE-processing-bakeHardening | 1 |
| FT-MSE-processing-coldDrawn | 1 |
| FT-MSE-processing-coldRolling | 1 |
| FT-MSE-processing-compressive-creep | 2 |
| FT-MSE-processing-deformation | 1 |
| FT-MSE-processing-drawing-cold | 1 |
| FT-MSE-processing-drawing-lowTemp | 1 |
| FT-MSE-processing-gasAtomisedPowder | 1 |
| FT-MSE-processing-heatTreatment | 1 |
| FT-MSE-processing-heatTreatment-T6 | 1 |
| FT-MSE-processing-heatTreatment-T6I6 | 1 |
| FT-MSE-processing-homogenised | 13 |
| FT-MSE-processing-hotIsostaticPressing | 3 |
| FT-MSE-processing-hotRolling | 2 |
| FT-MSE-processing-inductionMelting | 1 |
| FT-MSE-processing-ionIrradiation | 1 |
| FT-MSE-processing-magneticAnnealing | 1 |
| FT-MSE-processing-paintBaking | 1 |
| FT-MSE-processing-preaging | 1 |
| FT-MSE-processing-prestretched | 1 |
| FT-MSE-processing-recrystallised | 1 |
| FT-MSE-processing-remelting | 1 |
| FT-MSE-processing-rolled-cold | 1 |
| FT-MSE-processing-rolled-hot | 1 |
| FT-MSE-processing-selectiveLaserMelting | 1 |
| FT-MSE-processing-solutionHeatTreated | 2 |
| FT-MSE-processing-solutionisation | 2 |
| FT-MSE-processing-solutionTreated | 8 |
| FT-MSE-processing-stabilisation | 1 |
| FT-MSE-processing-strainHarden | 1 |
| FT-MSE-processing-tensile-creep | 1 |
| FT-MSE-processing-thermomechanicalTreatment | 1 |
| FT-MSE-processing-ultrasonicAdditiveManufacturing | 1 |

## Appendix A.3.5 Structure

Solid Solution

| | |
|---|---|
| FT-MSE-structure-solidSolution | 6 |
| FT-MSE-structure-soluteAggregate | 1 |
| FT-MSE-structure-solutePartitioning | 1 |
| FT-MSE-structure-SSSS | 1 |

Defects

| | |
|---|---|
| FT-MSE-structure-defects | 1 |
|     Cluster | |
| FT-MSE-structure-cluster | 6 |
| FT-MSE-structure-cluster-CuMg | 1 |
| FT-MSE-structure-cluster-MgAg | 1 |
| FT-MSE-structure-cluster-MgSi(Cu) | 1 |
| FT-MSE-structure-cluster-rich-Cu | 1 |
| FT-MSE-structure-cluster-rich-Mg | 1 |
| FT-MSE-structure-dispersoid | 2 |
| FT-MSE-structure-dispersoid-AlZr | 1 |
| FT-MSE-structure-GPBzone | 1 |
| FT-MSE-structure-GPzone | 6 |
| FT-MSE-structure-GPzone-enriched-Cu | 2 |
| FT-MSE-structure-GPzone-rod | 1 |
| FT-MSE-structure-GPzone-unitCell | 1 |
|     Precipitate | |
| FT-MSE-structure-bulk-precipitate | 1 |
| FT-MSE-structure-multiShell | 1 |
| FT-MSE-structure-phase-alpha1 | 1 |
| FT-MSE-structure-phase-alpha2 | 1 |
| FT-MSE-structure-phase-metastable | 1 |
| FT-MSE-structure-phase-Q | 1 |
| FT-MSE-structure-phase-stable | 1 |
| FT-MSE-structure-precipitate | 4 |
| FT-MSE-structure-precipitate-Al3Zr | 1 |
| FT-MSE-structure-precipitate-betaDoublePrime | 5 |
| FT-MSE-structure-precipitate-betaDoublePrime-LDC | 1 |
| FT-MSE-structure-precipitate-betaSn | 1 |
| FT-MSE-structure-precipitate-correlated-SnRich/CuRIch | 1 |
| FT-MSE-structure-precipitate-CunitCell | 1 |
| FT-MSE-structure-precipitate-disordered | 1 |
| FT-MSE-structure-precipitate-doubleShell | 3 |
| FT-MSE-structure-precipitate-earlyStage | 1 |
| FT-MSE-structure-precipitate-eta | 1 |
| FT-MSE-structure-precipitate-etaPrime | 1 |
| FT-MSE-structure-precipitate-nonUniformConcentration | 1 |
| FT-MSE-structure-precipitate-omega | 1 |
| FT-MSE-structure-precipitate-QP1 | 1 |
| FT-MSE-structure-precipitate-QP2 | 1 |
| FT-MSE-structure-precipitate-Qprime | 1 |
| FT-MSE-structure-precipitate-rich-Er | 1 |
| FT-MSE-structure-precipitate-rich-Zr | 1 |
| FT-MSE-structure-precipitate-shell | 1 |
| FT-MSE-structure-precipitate-thetaPrime | 3 |
| FT-MSE-structure-precipitate-typeFraction | 1 |
| FT-MSE-structure-precipitate-uniformConcentration | 1 |
|     Boundary | |
| FT-MSE-structure-grainBoundary | 2 |
| FT-MSE-structure-grainBoundary-precipitate | 1 |
| FT-MSE-structure-grainBoundary-precipitateFreeZone | 1 |
| FT-MSE-structure-grainBoundary-precipitates | 1 |
| FT-MSE-structure-grainBoundary-soluteSegregation | 1 |
| FT-MSE-structure-interface | 1 |
| FT-MSE-structure-interface-segregation | 1 |
| FT-MSE-structure-phaseBoundary-segregation-Y | 1 |

Microstructure

| | |
|---|---|
| FT-MSE-structure-phaseFraction | 1 |
| FT-MSE-eutectic-Si | 1 |
| FT-MSE-structure-ultraFineGrained | 1 |
| FT-MSE-structure-dendrite | 1 |
| FT-MSE-structure-lamellar | 1 |
| FT-MSE-structure-fractureSurface | 1 |

Appendix A.3.6 Property

Creep

| | |
|---|---|
| FT-MSE-property-coarseningResistance | 1 |
| FT-MSE-property-creep | 1 |
| FT-MSE-property-creepDuctility | 1 |
| FT-MSE-property-creepResistance | 2 |

Functional

| | |
|---|---|
| FT-MSE-property-conductivity | 1 |
| FT-MSE-property-electricalConductivity | 2 |
| FT-MSE-property-electricalResistivity | 1 |
| FT-MSE-property-magnetic | 1 |
| FT-MSE-property-ThermoElectricPower | 1 |

Corrosion

| | |
|---|---|
| FT-MSE-property-corrosionResistance | 1 |
| FT-MSE-property-stressCorrosionCrackingResistance | 1 |

Hardness

| | |
|---|---|
| FT-MSE-property-hardness | 1 |
| FT-MSE-property-hardness-micro | 1 |
| FT-MSE-property-temperHardness | 1 |
| FT-MSE-property-mechanical | 1 |
| FT-MSE-property-tensile | 6 |
| FT-MSE-property-tensile-UTS | 1 |
| FT-MSE-property-VickersHardness | 6 |
| FT-MSE-property-VickersHardness-micro | 7 |
| FT-MSE-property-yieldStrength | 2 |

Miscellaneous

| | |
|---|---|
| FT-MSE-property-elongation | 1 |

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
