# Peer review of "Atom Probe Tomography of Aluminium Alloys: A Systematic Meta-Analysis Review of 2018"

_metals, doi:10.3390/met9101071_

Round 1

Reviewer 1 Report

This review analyzes 34 very specific papers on aluminum alloys studied by APT. The PRISMA method is used to perform a systematic review. It is novel to use this method to analyze APT papers and focuses the materials science questions in alumnium alloys.

The article is a good review for the Special Issue "Application of Atom Probe Tomography in Metallic Materials".

There is only one misprint on line 22 (a double point) that should be removed.

Author Response

Thank you for your review. The typographical error has been fixed.

Reviewer 2 Report

The paper comprises a meta-study of APT of Al-alloys in 2018. The paper is very clear and I only have a few remarks.

1) I think it would be good to include some references to other meta-studies regarding the methodology used. Now only ref 35 is related to methodology, whereas 1-34 are the actual papers making the basis for the meta-study.

2) On row 113 it says "structure of particles". What is meant? I guess it is not the crystallographic structure of particles that is determined using APT, so is it the size, shape, composition, etc. that is meant? This could be clarified.

3) The division of instruments into laser/voltage instruments is not very useful, as presumably most of the studies used the voltage mode even if it was a laser instrument, and a laser instrument used in the voltage mode is more or less identical to a voltage instrument. However, I guess this issue does not necessarily need any modification of the paper. It would have been more interesting, though, to know which mode was actually used, but maybe it is too time-consuming to find out, or the information might not always be there.

Author Response

Thank you for your review. Here are our responses:

1) Only the PRISMA reference taken from the MDPI website was used because it was the only reference used in the creation of this work. We feel that adding other references would be inappropriate.

2) We have added the following sentence in line 117 for clarification:

"The full list of specific keywords for each type of question are listed in Appendix A.3.2."

3) Unfortunately, this distinction was not thought of until the manuscript was being written and the tools are not to a level of functionality where it is easy to go back a step and make changes. We did, however, already note this aspect in discussion in section 4.3:

"Note that this study did not effectively capture which mode was being used (either laser or voltage), only what type of instrument was being used"